# A New Approach to Obtaining Nano-Sized Graphene Oxide for Biomedical Applications

**DOI:** 10.3390/ma14061327

**Published:** 2021-03-10

**Authors:** Paulina Bolibok, Bartosz Szymczak, Katarzyna Roszek, Artur P. Terzyk, Marek Wiśniewski

**Affiliations:** 1Physicochemistry of Carbon Materials Research Group, Faculty of Chemistry, Nicolaus Copernicus University in Toruń, Gagarina 7, 87-100 Toruń, Poland; pbolibok@umk.pl (P.B.); aterzyk@chem.uni.torun.pl (A.P.T.); 2Department of Biochemistry, Faculty of Biological and Veterinary Sciences, Nicolaus Copernicus University in Toruń, Lwowska 1, 87-100 Toruń, Poland; b_szymczak@doktorant.umk.pl (B.S.); kroszek@umk.pl (K.R.)

**Keywords:** nanographene oxide, dimethyl sulfoxide (DMSO), ultrasonication, cytotoxicity, aggregation, capillary vessels

## Abstract

Graphene oxide (GO) is one of the most exciting and widely used materials. A new method of nanographene oxide (n-GO) formation is presented. The described unique sequence of ultrasonication in dimethyl sulfoxide solution allows us to obtain different sizes of n-GO sheets by controlling the timing of the cutting and re-aggregation processes. The obtained n-GO exhibits only minor spectral changes, mainly due to the formation of S-containing surface groups; thus, it can be concluded that the material is not reduced during the process. Maintaining the initial oxygen functionalities together with the required nano-size (down to 200 nm) and high homogeneity are beneficial for extensive applications of n-GO. Moreover, we prove that the obtained material is evidently biocompatible. The calculated half-maximal effective concentration (EC50) increases by 5-fold, i.e., from 50 to 250 µg/mL, when GO is converted to n-GO. As a consequence, the new n-GO neither disturbs blood flow even in the narrowest capillaries nor triggers a toxic influence in surrounding cells. Thus, it can be a serious candidate for drugs and biomolecule carriers administered systemically.

## 1. Introduction

Graphene oxide (GO) is one of the most exciting materials for the scientific community. It has been already synthesized for many years by numerous different procedures, e.g., Brodie’s [1], Staudenmaier’s [2], Hoffman’s [3], and Hummers’ [4,5,6] methods. Each one of these methods is based on graphite exfoliation in strong oxidative and acidic environment.

GO has remarkable and unique physicochemical properties. Oxygen-containing groups in conjunction with a large specific surface area and low toxicity have attracted growing interest and has offered promising applications in the biomedical field [7,8,9], including protein immobilization [10,11,12] and drug delivery [7,13].

Many features, underlain by the synthesis method, such as size or surface chemistry, can strongly affect GO biological properties. Thus, many researchers, using their own methods or modifications, have investigated GO biocompatibility [14]. The obtained data, both in vitro and in vivo, often show contradictory results that should be clarified. Hence, the determination of GO safety for living cells and organisms is still a challenge. Additionally, in most cases of biomedical application, material injection into the bloodstream and its distribution throughout the body should be considered. Thus, the systemic administration of GO sets novel requirements to be met. Large GO sheets introduced into the vascular system may clog capillaries, of which the internal diameter is less than 10 μm [15], and other blood vessels. Therefore, GO sheets used in biomedicine should be smaller; nevertheless, such “down-sizing race” cannot be unlimited. Too small GO pieces may damage the cell membranes, cross the membrane barrier, and undesirably penetrate the cells [16,17].

As mentioned above, numerous GO properties tailored for biomedical applications depend on its size and surface [18]. Considering the great urgency to find a method of fully size-controlled n-GO preparation, there are still attempts made in this area and further developments needed [19]. Many authors have prepared smaller and smaller materials in the form of n-GO, nanographene, and graphene quantum dots synthesized by different methods, among which two main groups (with and without sonication) can be indicated. Dos Santos and coworkers [20] described a four-step nanographene oxide synthesis. The carboxylated material obtained during the first three steps of synthesis was exposed to ultrasonication for 4 h in order to reduce the size of the flakes. Finally, carboxylated n-GO, sized 103.0 ± 0.5 nm based on dynamic light scattering (DLS) measurements, was prepared; however, the SEM results show the presence of small n-GO materials distributed among the large GO sheets [20]. The n-GO was then coupled with methylene blue (n-GO-MB) to create an active platform for the treatment of breast cancer in a murine model. Recently, nanographene oxide with a very wide size distribution from 50 to 500 nm was synthesized by Zhang et al. [21]. The authors obtained GO using an improved Hummers’ method and next reduced GO to graphene by adding sodium citrate. The dry material was ground into the powder and analyzed in terms of molecule absorption. The obtained material had the ability to absorb more methylene blue than graphite under the same conditions. In contrast, Mendes et al. [18] obtained two different sizes of GO nanosheets based on a modified Hummers’ method. In this case, the common top-down method was used together with sodium chloride as a crushing agent to exfoliate and to reduce the size of graphite.

During the complex and multistep process reported by Mendes et al. [18], including high temperature (100 °C) treatment and 3 h of sonication, the obtained material was partly reduced. Next, n-GO was exposed to sonication for 30 min or 4 h. It allowed them to obtain different sizes of n-GO (mean diameter of 89 and 277 nm, based on atomic force microscopy (AFM) analysis) and to compare them in terms of cytotoxicity and uptake into cells. The authors concluded that the smaller n-GO sheets are more biofriendly.

An interesting approach to n-GO preparation by using graphite nanofibers as a starting material for a multistep process was described by Luo et al. [22]. In the first step, graphite nanofibers were exposed to pre-oxidative conditions. After that, a modified Hummers’ method was applied. The authors suggested that the diameter of the obtained material is determined by both the starting material diameter and the oxidation time. Moreover, they showed that n-GO colloids were more homogenous and stable than a material prepared from graphite powder. The stability of n-GO seems to be another important feature promoting its widespread application.

Interestingly, not only GO is a substrate in nano-sheets formation. Adel et al. [23] showed facile synthesis of nanographene layers using an expensive, supercritical phase exfoliation method combined with an ultrasonication process. Different synthesis conditions (i.e., pressure, sonication time, sonication amplitude, and amount of starting graphite) were tested. By using the most optimal conditions, the synthesis of nanographene with an average size of 128 nm has been described [23]. However, one of the most important disadvantages of this method is the equipment used for synthesis.

In contrast to the abovementioned multistep and time-consuming methods, we report the fast, simple, and cheap preparation of a size-controlled n-GO method. The applied-for-the-first-time post-synthesis parameters, i.e., DMSO/H_2_O mixture and unique ultrasonication sequence, allowed us to obtain n-GO with very narrow size distribution and limited number of layers. Such a size-controlled n-GO fulfills all requirements for biomedical applications including drug delivery and systemic administration of nanomaterial. The obtained n-GO is nontoxic and can easily pass through narrow capillaries without doubts about clogging.

## 2. Materials and Methods

A modified Hummers’ method was used to prepare graphene oxide (GO) [24]. Three grams of graphite flakes were added to a mixture of concentrated H_2_SO_4_/H_3_PO_4_ (360:40 mL) and stirred in a flask for 30 min. Then, KMnO_4_ (18 g) was slowly added to the mixture, producing slightly exothermic conditions (35–40 °C), and mixed for 24 h. After, 400 mL of water and 5 mL of 30% H_2_O_2_ were added, producing a bright yellow sol. The obtained mixture was centrifuged at 10,000 rpm for 30 min, and the supernatant was separated. The recovered material was washed with water and ethanol and centrifuged repeatedly (several times) to adjust the pH to ≈5. Subsequently, the obtained GO was exposed to ultrasonication in DMSO/H_2_O mixture. In brief, the appropriate dose (2.90 mL) of GO water suspension (1.65 mg/mL) was added to glass bottles containing 5.1 mL of pure DMSO. Next, the mixture was treated with ultrasounds in ice baths for different times (from 1 h to 68 h) according to the time sequence shown in Figure 1. The final solid Xn-GO (where X represents the time of ultrasonication ranging from 0 for pristine GO to 68 h) was obtained by drying it under vacuum for further use. Note that the obtained material was easily re-dispersed in water.

The obtained nanographene oxide sheets and pristine graphene oxide solution were analyzed by high-resolution transmission electron microscopy (HRTEM). The images were taken using a transmission electron microscope F20X-TWIN (FEI-Tecnai, Norcross, GA, USA) operated at 200 kV. Tapping mode atomic force microscopy (AFM) measurements were performed using a Veeco microscope (Veeco Metrology, Santa Barbara, CA, USA) with an NSG-11 probe (scan size 2–10 μm; scan rate 1 Hz, tapping mode). The Xn-GO and initial GO samples for HRTEM and AFM were prepared by dropping the solutions onto silica wafer (AFM) or Lacey Carbon film on Copper 400 mesh (HRTEM).

The Fourier transform infrared (FTIR) measurements spectra were accomplished by Mattson Genesis II infrared spectrophotometer (Mattson, Foster City, CA, USA) using transmission mode techniques in the frequency range 400–6000 cm^−1^.

The hydrodynamic diameter was evaluated by dynamic light scattering (DLS), using Particulate Systems, NanoPlus HD (Micromeritics, Particulate Systems, Norcross, GA, USA). All measurements were carried out at 25 °C.

Additionally, nanocolloidal stability tests were performed by registering UV-vis spectra (Jasco 650, Jasco Int. Co., Tokyo, Japan). The samples were centrifuged at different forces (600–10,000× *g*). Furthermore, Xn-GO (X = 1, 2, and 3) were centrifuged at 10,000× *g* for 20–150 min.

For the Raman measurements, the nonpolarized spectra of the structures were investigated in the spectral range of 60−4500 cm^−1^. The Raman spectra were recorded in the backscattering geometry using a SENTERRA micro-Raman system (Bruker Optik, Billerica, MA, USA). The green laser operating at 532 nm was used as an excitation light. The laser beam was tightly focused on the sample surface through a 20× microscope objective. To prevent any damage of the sample, an excitation power was fixed at 0.2 mW. A CCD (Charged Coupled Device) camera with temperature of 223 K, a laser spot of about 5 μm, and a total integration time of 100 s (50 × 2 s) were used. The position of the microscope objective in relation to the sample was piezoelectrically controlled (XY position).

The size-dependent cytotoxicity of the pristine material and Xn-GO sheets was investigated using an in vitro model. The human dermal fibroblast (HDF) cells were purchased from Biokom. The cells were grown in DMEM-LG (Dulbecco’s Modified Eagle’s Medium, Low Glucose) medium containing 10% FBS (Fetal Bovine Serum), 1% antibiotics, in a CO_2_ incubator at 37 °C in humidified atmosphere containing 5% CO_2_. A volume of 5 μL containing approximately 1 × 10^4^ cells was seeded to each well of a 96-well plate 24 h before the experiment started. GO and 2n-GO were added to the growing cells in concentrations of 1, 10, 50, 100, 250, 500, and 750 μg/mL and incubated for the next 24, 48, and 72 h. Subsequently, an MTT (3-(4,5-dimethylthiazol-2-yl)-2,5-diphenyltetrazolium bromide test based on the ability to reduce (MTT) by mitochondrial dehydrogenases and LDH (lactate dehydrogenase) assay determining the cell membrane integrity were performed in triplicate to assess the cell metabolic activity and viability, respectively.

The potential for biomedical applications of 2n-GO samples was also determined as the ability of materials to flow through capillaries and other blood vessels. A nylon filter membrane with pore size of 10 µm was used as a capillary model. After each single filtration, a UV-vis spectrum was taken to compare retaining GO on a filter membrane.

## 3. Results and Discussion

The DLS analysis results (Figure 2A) were the first criterion defining the best conditions of Xn-GO (where X represents the time of ultrasonication ranging from 0 for pristine GO to 68 h) synthesis. The pristine material contained very large and size-distributed sheets mainly observable as two DLS signals: a smaller one at 9872.4 ± 3484.5 nm and a larger one at 48,662.6 ± 22,836.8 nm.

The structure decreases in size under sonication in the DMSO/H_2_O mixture very quickly. After a 1-h exposure to ultrasounds sheets of 513.7 ± 221.1 nm was obtained and after 2-h exposure, the measured size was 231.4 ± 14.8 nm. It is extremely important to note that, beside the decrease in the mean diameter, the size distribution also narrowed. However, the further prolonged ultrasonication time (to 3, 7, and 68 h) resulted in obtaining larger GO sheets, and more heterogeneous and more unstable nanocolloids (Figure 2B). Based on these results, we rejected the 68n-GO sample as the sonication treatment over 7 h did not cause any beneficial alterations. The other samples were compared in terms of nanocolloidal stability, which increases gradually for the 1n-GO, 2n-GO, and 3n-GO samples. The fact that the 7n-GO sample is even less stable than 0n-GO (Figure 2(C1)) drove us to reject the sample. The most stable nanocolloid was formed by the 2n-GO sample.

Using the reason that similar nanolayer sizes after 2-h and 3-h ultrasonication were achieved, both colloids’ stabilities were compared by suspension storage at room temperature for 14 days (Figure 2D). The DLS measured size increased for both samples; however, for 3n-GO, the stability in water was lower when compared to the 2n-GO sample. This effect may be due to aggregation of the 3n-GO sheets consequent from increases in π–π stacking (Figure 2D). Thus, the best ultrasonication time for sample modifications was experimentally established as 2 h (2n-GO sample).

The Raman spectra of the obtained Xn-GO series were deconvoluted using Gaussian curves (Figure 3). The perfect fits with Gaussian band shapes reflect inhomogeneous broadening and are the consequences of the existence of enough long-living species in the tested systems. The spectra consist of the first-order Raman modes, namely, the D_4_, D, D_3_, G, and D’ bands. The most intense—D band, which is located near 1350 cm^−1^—is connected to sp^3^ carbons and arises usually from the defects and disorders in the carbon lattice [25].

The G band with a maximum at ca. 1580 cm^−1^ corresponds to the Raman-allowed E_2g_ optical phonon [26] and is a typical feature of all graphitic materials. The deconvolution revealed the presence of three other signals of different symmetries. The blue shoulder of the G-band–D′ signal appearing at ca. 1610 cm^−1^, the wide band at 1450–1500 cm^−1^—the D_3_ band—and the D_4_ band located at ca. 1280 cm^−1^ are related to the amorphous phase [27,28,29].

Our recent results (data under review) show that the D_4_ peak intensity is highly correlated to the folding of a GO sheet and, thus, to the presence of long range surface defects. By “long range”, we mean the curvature radius of dozens of nanometers. On the other hand, D’ is assumed as the factor of “short-range defects”. Moreover, based on the origins of the D and G bands, the I_D_/I_G_ ratio is usually applied as a quantitative index of the defects in carbon materials [30,31,32]. The comparative spectroscopic analysis of GO during the size reduction process shows significant changes in Raman peak intensities and their ratio. While the intensity of the D band becomes constant during the process, the changes in I_D_/I_G_ are connected to the I_G_ increase, which is followed by stretching the folds and disappearing of wrinkles. All these changes are observable in the AFM and HRTEM analyses, as shown in Figure 4.

The HRTEM and AFM methods were applied to monitor the changes in morphology and size of Xn-GO (X from 0 to 68) during each synthesis method. Although HRTEM and AFM pictures show only a part of the sample, presented in Figure 4, the 3D AFM images confirm the successful formation of nanosheets (“cutting” of GO sheets to nano-sized layers). The 2n-GO sample contains sheets with a mean diameter 221 ± 85 nm and ≈5 layers (Figure 4B). In comparison to 2n-GO sample, the 3n-GO sample (Figure 4C) size was approximately 301 ± 92 nm and ≈11 layers [18].

The number of layers increases with the rise in ultrasonication time similar to that observed by Mendes et al. [18] The authors obtained 4- and 6-layer structures respectively after 0.5 and 4 h of continuous sonication processes. However, their results pointed also to advanced reduction in the surface oxygen functionalities.

In our case, the increase in layer number seems to be faster and more effective. By comparing the results, one can conclude that π–π interactions are less effective than dipole–dipole ones present in the highly functionalized samples.

Regarding size, the result corresponds with those obtained by DLS analysis. Figure 4B’ shows that the obtained flakes of the 2n-GO sample have neither folds nor wrinkles when compared to native material (Figure 4A’). The HRTEM analysis confirms the AFM results, suggesting a flexible structure of 2n-GO. A further increase in the time of sonication, i.e., in the case of the 3n-GO sample (Figure 4C’), causes a wrinkled and folded morphology to appear, presumably due to aggregation, similar to that for raw material.

Therefore, we can conclude that, while GO sheets become smaller, their surface becomes flatter (D_4_ intensity decrease—Figure 3B) and the character of the defects becomes more local (intensity of D’ increase). Thus, the apparent increase in graphitization (I_D_/I_G_ decreasing) is ascribed to the unfolding process.

Based on the above outcomes, we chose the 2n-GO sample as the best candidate for further investigations. At first, we checked the changes in surface chemistry.

By comparison of the spectra collected in Figure 5A, one can conclude that the process of size reduction does not cause any drastic changes in surface chemistry. In the IR spectrum of the pristine 0n-GO sample, four main signals (1740, 1600, 1420, and 1100 cm^−1^) in the fingerprint region are present. The band at 1740 cm^−1^ is attributed to carbonyl compounds. The low intensity as well as the signal width evidence the spread in different chemical vicinities. Similarly, the ν(C=C) band, appearing as a small and overlapped signal near 1600 cm^−1^, means that the graphene sheet is fully covered with O-atoms. The latter implication is confirmed by the presence of two other intense and narrow bands at 1420 and 1100 cm^−1^.

The 2-h sonication process causes slight spectral changes (Figure 5A(b)), which are visible only in the differential spectrum (Figure 5A(c)) and are underpinned by the formation of S-containing surface groups such as S=O (–SO_2_ and/or –SO_3_) and C=S. Their amount, as it was confirmed by EDX investigations, is small, i.e., 3–5 at%, which is in agreement with the literature data [33,34]. Nevertheless, the increase in S content in the 2n-GO sample is statistically important. This also means that the applied unique sonication sequence and DMSO/H_2_O mixture prevent a reduction in GO during “cutting”, and that is in contrast to the results presented in the literature, e.g., [18]. The reduction is linked mainly with sample overheating and causes the increase in graphitization degree. The latter is often described as the main basis of material cytotoxicity.

The toxicity of the raw 0n-GO material and 2n-GO sheets was investigated to confirm the hypothesis of biocompatibility enhancement after down-sizing the GO sheets. The HDF cell line was cultured with sample suspensions in seven different concentrations ranging from 1 to 750 μg/mL for three select time periods (24, 48, and 72 h). In order to compare toxicity, standard MTT and LDH assays were used. The cell mitochondrial activity (viability) assayed with an MTT test is concentration- and time-dependent (Figure 6A–C). After the first 24 h of incubation (Figure 6A) with the materials, the cell viability after exposure to 2n-GO solution is higher than after exposure to 0n-GO. This is especially noticeable at high material concentrations exceeding 50 μg/mL. The data obtained after 48 h (Figure 6B) and 72 h (Figure 6C) exposure also reflect these relations. The 2n-GO toxicity after 72 h is significantly lower than 0n-GO, with EC50 values of 250 and 50 μg/mL, respectively.

These results suggest that larger flakes are generally more toxic. The optical microscopic images of growing cells (Figure 6E) also confirmed that 0n-GO in the form of large sheets covered the cells and therefore contributed to toxic influence. On the other hand, the obtained 2n-GO sample is more biocompatible even at higher concentrations and seems to be less toxic than other n-GO described in the literature [18,35].

Smaller n-GO flakes pose a risk of gradual cell membrane damage. In order to check the cell membrane integrity, the LDH activity (Figure 6D) was assayed. The LDH activity in the culture medium increases with increasing cell damage. The obtained data suggest no considerable damage triggered by either 0n-GO or 2n-GO on cell membranes. Larger GO sheets remain outside the cells and occasionally can induce some disturbances, while 2n-GO flakes are smaller fragments that do not penetrate and damage the cell membrane. These observations are confirmed by microscopy images (Figure 6E).

As mentioned in Section 1, GO could be used as a carrier for drugs and other bioactive substances in systemic administration, only if its size is small enough to prevent vessel clogging and simultaneously not too small to avoid crossing the cell barrier. Thus, we checked the possibility of 2n-GO and 0n-GO flow through the membrane used as a model of capillary flow. The analysis of tested material solutions included eight cycles for each sample flowing through nylon filter membrane with pore sizes of 10 µm. The UV-Vis spectra shown in Figure 7 confirm that 2n-GO has the ability to be obtained through the narrow space without a concentration decrease. In contrast to the 2n-GO solution, the amount of flowing 0n-GO decreased gradually with each cycle.

The pristine material, based on the results presented in Figure 2, consists of GO sheets ranging from 1 to 100 µm. The observed aggregation process (and potential clogging of the capillaries) is progressive and reaches over 95% of GO uptake already after 6 cycles.

In this study, we show that the obtained nano-sized material has the same spectral and chemical properties as pristine GO. This means that the application of a unique sonication sequence lets us prevent sample overheating and the subsequent reduction in surface oxygen functionalities. It is the key feature as the material surface chemistry and size underpin the interactions with cells. In the case of GO-derived materials, it is commonly accepted that lower graphitization degrees contribute to lower cytotoxicity. Thus, one can conclude that 2n-GO due to a smaller size and more favorable chemistry of the surface is nontoxic, is not able to aggregate, and therefore will not clog the capillaries after systemic administration.

Moreover, the comparison of different preparation methods for nano-sized graphene oxide (summarized in Table 1) points out the benefits of the procedure described in this work.

## 4. Conclusions

To the best of our knowledge, the method of n-GO production described in this study is the first procedure where concerted action of DMSO and an ultrasonication unique sequence was used. The obtained 2n-GO is highly homogenous, which is confirmed by DLS analysis. Moreover, 2n-GO has not been reduced during preparation, which is superior over the materials obtained by others. The smaller material is evidently more biocompatible than pristine 0n-GO. It can be a consequence of controlled size-reduction and retaining the oxygen surface functionalities, formed during GO synthesis. Nevertheless, the obtained results indicate that 2n-GO can be used as a carrier of biomolecules and drugs administered systemically. We are actively pursuing this issue, and detailed studies on the use of 2n-GO in this field will be presented shortly.

## Figures and Tables

**Figure 1 materials-14-01327-f001:**
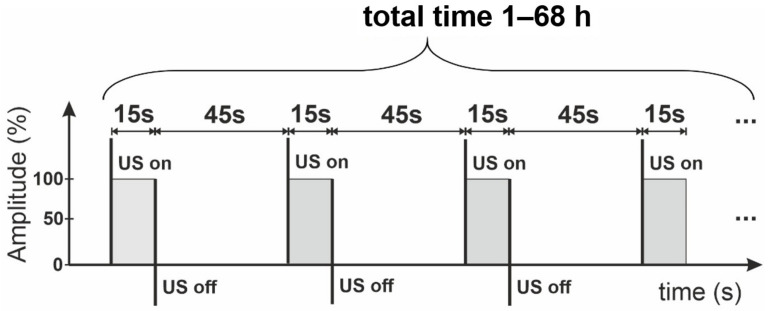
The sequence of ultrasonication (US) used for Xn-graphene oxide (GO) (X = 0, 1, 2, 3, 7, and 68) production. Note that the 15 s working intervals are followed by 45 s delays; X—the time of sonication represents the total proceeding time.

**Figure 2 materials-14-01327-f002:**
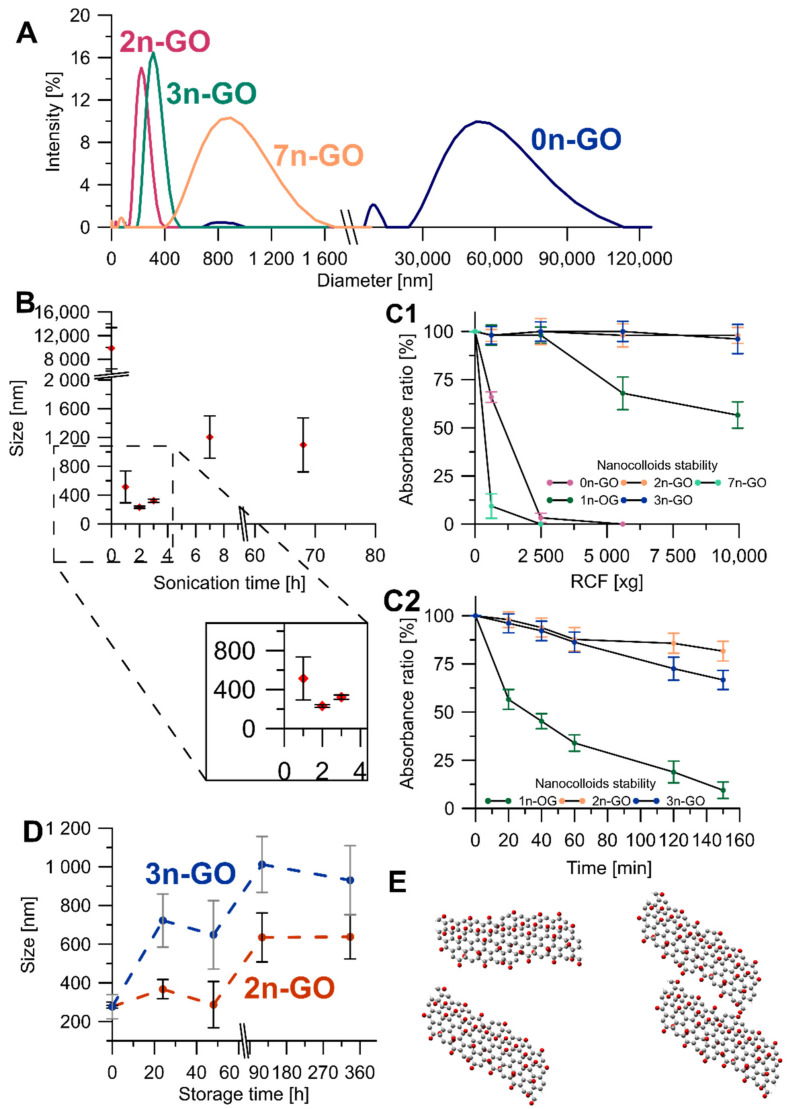
(**A**) The Xn-GO size distribution obtained from dynamic light scattering (DLS) analyses; (**B**) the Xn-GO size as a function of ultrasonication time; (**C**) nanocolloids stability as a function of centrifugation 1— force and 2—time; (**D**) comparison of 2n-GO and 3n-GO nanocolloid’s time stabilities (note that the dashed lines are for visualization only); (**E**) model of a successive aggregation process.

**Figure 3 materials-14-01327-f003:**
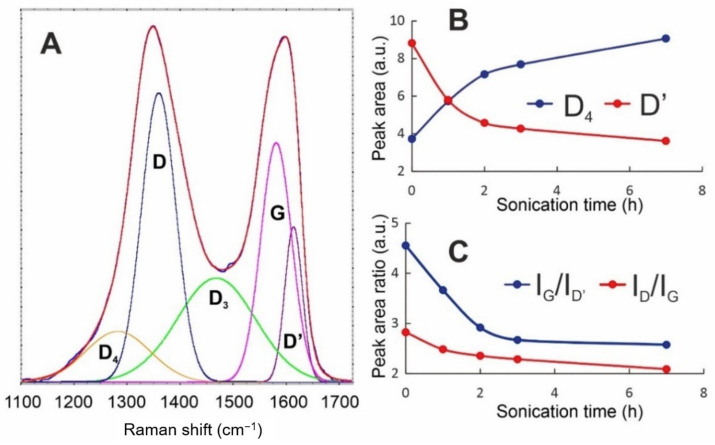
Raman spectroscopic analyses of the size reduction process: (**A**) deconvoluted spectrum of 0n-GO (showing the fitted model); (**B**) the intensity of D_4_ and D’ changes during the process; (**C**) effect of the sonication time on the I_G_/I_D’_ and I_D_/I_G_ ratios.

**Figure 4 materials-14-01327-f004:**
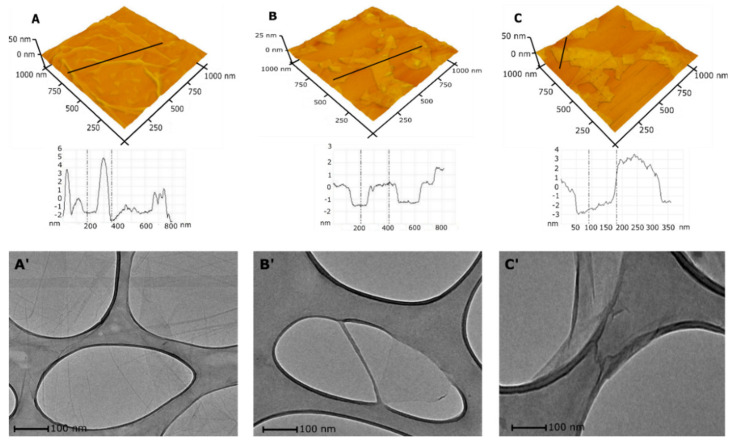
The results from microscopic analyses indicating morphological changes during the sonication process. Upper panel: 3D atomic force microscopy (AFM) images of pristine GO, (**A**) 2n-GO (after 2 h sonication), (**B**) and 3n-GO (**C**). The graphs in the middle show the corresponding cross sections. The bottom panel: high-resolution transmission electron microscopy (HRTEM) images of pristine GO material (**A’**), 2n-GO (**B’**), and 3n-GO (**C’**).

**Figure 5 materials-14-01327-f005:**
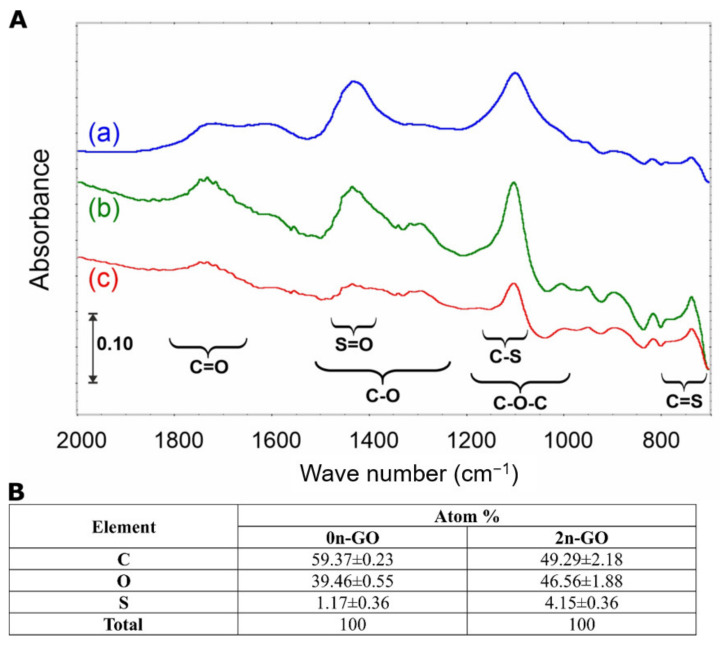
(**A**) Fourier transform infrared (FTIR) spectral changes registered during a 2-h size reduction process: (a) initial GO sample, (b) 2n-GO, and (c) differential spectrum. (**B**) The results from EDX measurements. Data are presented as the mean ± standard error of the mean (SEM). All differences in the elemental composition were statistically important with *p* < 0.001.

**Figure 6 materials-14-01327-f006:**
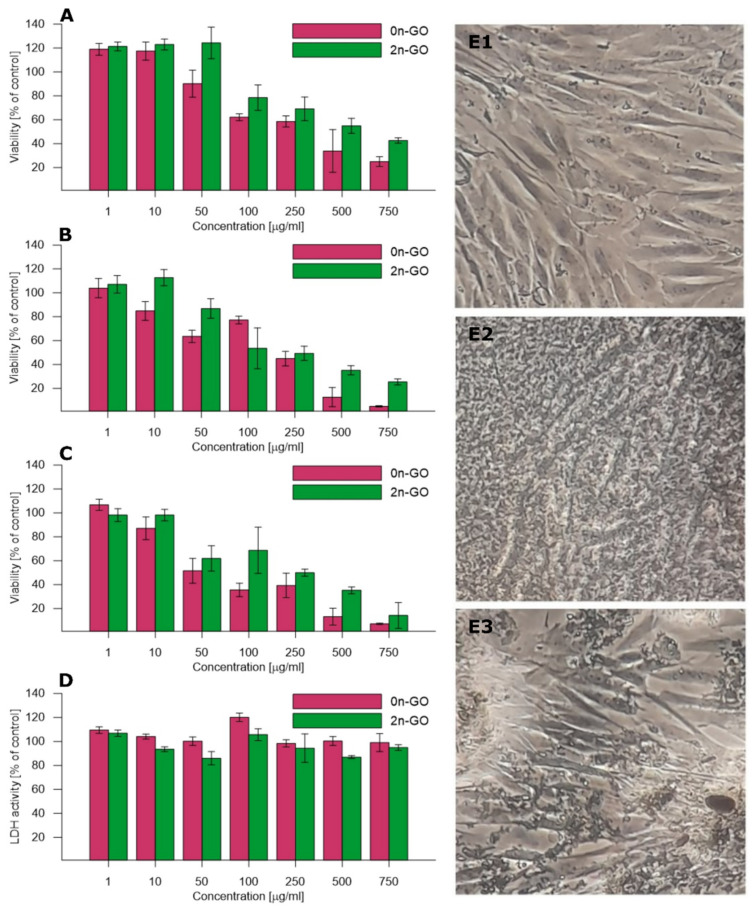
Comparison of 0n- and 2n-GO toxicity towards human dermal fibroblasts: results of the MTT test after (**A**) 24, (**B**) 48, and (**C**) 72 h, respectively (cell viability related to control cells without GO supplementation); (**D**) LDH activity (related to control cells without GO supplementation). Optical microscopic images of HDF cell line: (**E**) 1—control cells, 2—cells cultured with 0n-GO solution, and 3—cells cultured with 2n-GO solution.

**Figure 7 materials-14-01327-f007:**
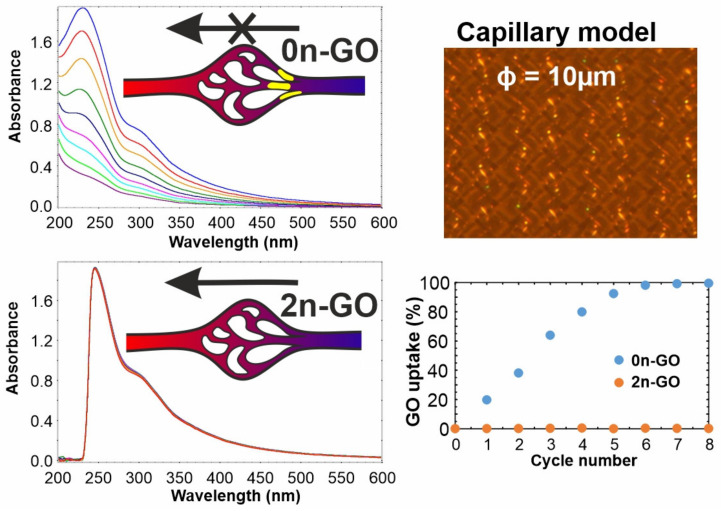
UV-Vis spectra of 2n-GO and 0n-GO down-flow. 0n-GO concentration decreases during the experiment, whereas the 2n-GO amount in solution after eight flow cycles was not changed.

**Table 1 materials-14-01327-t001:** Summary of n-GO synthesis methods proposed in the literature indicating the differences in selected nano-sized graphene oxide properties.

Source Material (Method)	Number of Layers	Size (nm)	Stability	Surfaces Chemistry	Number of Steps (Time Consuming)	References
Graphite flakes (us 30 min; or 4 h)	4 6	277 89	n.a.	Reduced	2 (++)	[18]
Graphite flakes (us 4 h + cf-100 kDa)	n.a.	103.0 ± 0.5	+	Reduced and Carboxylated	4 (+++)	[20]
Graphite powder (no us)	n.a.	50–500	n.a.	Reduced	12 (+++)	[21]
Graphite nanofibers (no us)	n.a.	18 46	+++	Oxidized	2 (++)	[22]
Graphite flakes (us 5/5 min)	1/2	20–80 30–200	+	Partially reduced	2 (+)	[35]
Graphite flakes (us 15/45 s)	5	231.4 ± 14.8	+++	Oxidized	2 (+)	This paper

Stability/time consuming: + low/not meaningful; ++ average; +++ high/meaningful; us—ultrasonication; cf—centrifuge filtration.

## Data Availability

Data sharing not applicable.

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
