# Peer review of "A New Approach to Obtaining Nano-Sized Graphene Oxide for Biomedical Applications"

_materials, 2021, doi:10.3390/ma14061327_

Round 1
Reviewer 1 Report
Comments:
The manuscript entitled with “A new approach to obtaining nano-sized graphene oxide for biomedical applications”describes sonication methods to modulate the size of graphene oxide for the purpose of biomedical applications. The manuscript can be considered for "Materials" after below minor revision.
- Please discuss why the sonication treatment over 7 h incurs larger GO sheets.
- Zhang et al. described the method to make GO size from 50 to 500 nm. Please discuss how the authors' method can be applied to make GO to be smaller than 200 and 100 nm.
- Please perform Figure 5B (table) at least three times in order to compare 0n-GO and 2n-GO statistically (mean +- Standard deviation & Student t test). It is unclear whether the increase of thiol contents in 2n-GO is scientifically meaningful. In addition, if the increase of thiol contents in 2n-GO is statistically meaningful, please discuss how thiol contents are increased during sonication; where this increased thiol comes from?.
- Please re-arrange the Figure 6A and 6B to compare the 0n-GO and 2n-GO directly.
- Please conduct the colloidal stability of Xn-GO, which may emphasize the importance of size-reducing in biomedical applications of GO
- Please define the abbreviation when it is first appeared; eg. MTT, LDH, etc.
- What are the blue and orange dots in the right bottom graphs in Figure 7?
- Please describe the method and logics of experiments for Figure 7 in the results sections in more detail.
- Please describe the figures in more detail in all figure legends.
Reviewer 2 Report
Bolibok et al. present a very interesting study about a new approach for obtaining nanosized graphene oxide particles for biomedical applications.
The paper clearly presents the advantages and drawbacks of the routes employed in the literature and reports a fast, simple and cheap preparation of size-controlled graphene oxide. The authors provide extensive physicochemical characterization of the n-GO by employing several techniques (Raman, HRTEM, AFM, FTIR, etc.). Toxicity assessments were also performed to establish the biocompatibility of these materials.
I find these results particularly interesting for the readers of Materials and I have no major comment on the work performed. In my opinion the manuscript can be accepted in its present state.
Reviewer 3 Report
Major revision is required. The major concern is the following point,
“Described unique sequence of ultra-12 sonication in dimethyl sulfoxide solution allows to obtain desirable size of n-GO sheets through full 13 control of the cutting and re-aggregation processes….The maintaining of the initial oxygen functionalities, 16 together with tailored size and high homogeneity are beneficial for extensive applications of n-GO.”
In fact, what reported here does not support above claim (tailored size, desirable size, full control).
2n-GO appears to be the minimum sized one.
“where X 112 represents the time of ultrasonication ranging from 0 for pristine GO to 68h”
Only limited results are reported in the paper. Pls try to include all.
Fig 3(a). It is strange that the size of GO is increased in 7n-GO (in comparison with that of 2n-GO). If this really could happen, it raises a big question wrt quality control of the resulted GO, and need further parametric study to ensure the quality of the resulted GO.
The results reported here (e.g., Fig 2) might be dependent on the quality of the original raw material. However, currently, it is difficult to obtain a large quantity of such a raw material with stable quality. How are you go to deal with this?
“Based on the above outcomes, we have chosen the 2n-GO sample as the best candi-244 date for further investigations.”
Why 2n-GO is the best candidate? All or at least a few should be tested for comparison.
Additionally,
- A table to systematically compare different ways to prepare nano-sized graphene oxide should be included at the end of this paper, in terms of the cost, time consuming, quality, size range… etc.
- The exact meaning of Fig 1 is very hard to catch. The related context needs to improve as well.
- Storage time mentioned in Fig 3(c) does not seemly match that in Fig 1. Same for sonication time.
Round 2
Reviewer 3 Report
acceptable now